# Reactivation of Multiple Fetal miRNAs in Lung Adenocarcinoma

**DOI:** 10.3390/cancers13112686

**Published:** 2021-05-29

**Authors:** David E. Cohn, Mateus C. Barros-Filho, Brenda C. Minatel, Michelle E. Pewarchuk, Erin A. Marshall, Emily A. Vucic, Adam P. Sage, Nikita Telkar, Greg L. Stewart, Igor Jurisica, Patricia P. Reis, Wendy P. Robinson, Wan L. Lam

**Affiliations:** 1British Columbia Cancer Research Centre, Vancouver, BC V5Z 1L3, Canada; mbarros@bccrc.ca (M.C.B.-F.); bminatel@bccrc.ca (B.C.M.); mpewarchuk@bccrc.ca (M.E.P.); emarshall@bccrc.ca (E.A.M.); Emily.Vucic@nyulangone.org (E.A.V.); asage@bccrc.ca (A.P.S.); ntelkar@bccrc.ca (N.T.); gstewart@bccrc.ca (G.L.S.); wanlam@bccrc.ca (W.L.L.); 2International Research Center, A.C. Camargo Cancer Center, São Paulo, SP 01525-001, Brazil; 3NYU Langone Medical Center, New York, NY 10016, USA; 4British Columbia Children’s Hospital Research Institute, Vancouver, BC V5Z 4H4, Canada; wrobinson@bcchr.ca; 5Department of Medical Genetics, University of British Columbia, Vancouver, BC V6H 3N1, Canada; 6Osteoarthritis Research Program, Division of Orthopedic Surgery, Schroeder Arthritis Institute, University Health Network, Toronto, ON M5T 0S8, Canada; juris@ai.utoronto.ca; 7Data Science Discovery Centre for Chronic Diseases, Krembil Research Institute, University Health Network, Toronto, ON M5T 0S8, Canada; 8Department of Medical Biophysics, University of Toronto, Toronto, ON M5G 1L7, Canada; 9Department of Computer Science, University of Toronto, Toronto, ON M5S 2E4, Canada; 10Faculty of Medicine, São Paulo State University (UNESP), Botucatu, SP 18618-687, Brazil; patricia.reis@unesp.br

**Keywords:** lung adenocarcinoma, microRNA, novel microRNA, oncofetal

## Abstract

**Simple Summary:**

Patterns of microRNA expression in fetal tissues are not well-characterized, due to the rarity of human fetal samples. Characterization of these patterns is vital for improving our understanding of developmental disorders, and can also provide insights into cancer development, as tumours frequently exploit developmental pathways to facilitate their uncontrolled growth. To profile fetal microRNA expression, we compared the small RNA transcriptomes of a unique cohort of 25 fetal lung samples and two independent cohorts of adult lung specimens, each containing adenocarcinoma and non-malignant samples. We identified 13 ‘oncofetal’ microRNAs that were highly expressed in the fetal and adenocarcinoma samples but absent from the adult non-malignant samples. These microRNAs showed potential as markers for cancer detection, and the expression of three of them was associated with shorter survival times for lung adenocarcinoma patients. The absence of these microRNAs from the non-malignant adult lung also makes them compelling targets for novel therapies.

**Abstract:**

MicroRNAs (miRNAs) play vital roles in the regulation of normal developmental pathways. However, cancer cells can co-opt these miRNAs, and the pathways that they regulate, to drive pro-tumourigenic phenotypes. Characterization of the miRNA transcriptomes of fetal organs is essential for identifying these oncofetal miRNAs, but it has been limited by fetal sample availability. As oncofetal miRNAs are absent from healthy adult lungs, they represent ideal targets for developing diagnostic and therapeutic strategies. We conducted small RNA sequencing of a rare collection of 25 human fetal lung (FL) samples and compared them to two independent cohorts (*n* = 140, *n* = 427), each comprised of adult non-neoplastic lung (ANL) and lung adenocarcinoma (LUAD) samples. We identified 13 oncofetal miRNAs that were expressed in FL and LUAD but not in ANL. These oncofetal miRNAs are potential biomarkers for LUAD detection (AUC = 0.963). Five of these miRNAs are derived from the imprinted C14MC miRNA cluster at the 14q32 locus, which has been associated with cancer development and abnormal fetal and placental development. Additionally, we observed the pulmonary expression of 44 previously unannotated miRNAs. The sequencing of these fetal lung samples also provides a baseline resource against which aberrant samples can be compared.

## 1. Introduction

Healthy embryonic development requires rapid cell differentiation, proliferation, and migration. The genes responsible for these processes are typically inactive in adult tissues [1,2], which generally have slower rates of cell turnover. This inactivation occurs through a variety of mechanisms, including transcription factor binding, DNA methylation, and microRNA (miRNA) mediated mRNA degradation or translational inhibition [3,4,5]. In cancer, malignant cells commonly disrupt these regulatory mechanisms, allowing them to reactivate the expression of fetal genes and proteins and recapitulate the rapid cell division and migration seen during development [6,7]. These oncofetal proteins are so named because they are active during development and in malignant tumours, but not in non-malignant adult tissue. Their lack of expression in healthy adult tissue makes them compelling drug targets and biomarkers of disease [7,8].

Malignant tumours commonly alter the expression of miRNAs, which are key epigenetic regulators of developmental pathways. MiRNAs are small non-coding RNAs (ncRNAs) that are 19–25 nucleotides in length, and that primarily negatively regulate mRNA transcripts by binding their 3′ untranslated region. The miRNA transcriptome of most human tissues is highly diverse, with a total of 2654 mature human miRNAs being annotated in miRBase v22 [9]. In lung cancer, deregulation of miRNA expression is a common mechanism through which tumours modulate the activity of critical oncogenes and tumour suppressors, including *MYC* and *PTEN* [10,11]. This characteristic deregulation of miRNAs in lung cancer makes them useful biomarkers, both for detecting tumours and predicting the likelihood of progression, metastasis, or recurrence [12,13].

Several studies have shown specific examples of ncRNAs that follow an oncofetal expression pattern and promote pro-tumourigenic phenotypes in adults by directly or indirectly downregulating tumour-suppressive molecules [14,15]. However, the large-scale identification of oncofetal miRNAs in human cancer has been limited by the availability of fetal samples. Studies aiming to discover oncofetal miRNAs have typically employed very few fetal samples or used materials such as fetal cell lines that do not accurately reflect the biology of human fetal organs [15,16,17]. For example, the analysis of eight fetal colorectal specimens has led to the identification of miR-17-5p as an oncofetal miRNA, which represses its target gene *P130* in colorectal cancer [16]. In a previous study, we analyzed five fetal lung samples and found aberrant expression of oncofetal miRNAs that target the 3′-UTR of the transcription factor Nuclear Factor I/B (*NFIB*) mRNA, which is critical to lung development [15]. Together, these studies have succeeded in demonstrating the principle that the expression of certain developmentally-active miRNA genes is reactivated within adult tumours, but their small sample sizes prevent them from serving as representative resources for fetal miRNA expression.

Similar to oncofetal proteins, the absence of oncofetal miRNAs from healthy adult tissue makes them ideal markers for cancer detection, as well as targets for developing precision therapies. A high-resolution view of miRNA expression in the fetal lung is vital for the identification of oncofetal miRNAs and their target genes in human lung cancer. As such, there is a great need for a detailed characterization of the fetal lung miRNA transcriptome, based on a more comprehensive set of samples. This characterization will also provide a baseline against which the transcriptomes of individual fetal samples can be compared. MiRNAs are known to play key roles in developmental biology [18], and so this resource would enable the discovery of alterations that occur in aberrant lung development driven by genetic or environmental causes, facilitating the study of fetal developmental disorders, such as congenital lung malformations.

Here, we present a characterization of the fetal lung (FL) miRNA transcriptome based on 25 human fetal lung samples and draw comparisons to the miRNA transcriptomes of the adult non-neoplastic lung (ANL) and lung adenocarcinoma (LUAD). Based on this characterization, we identified a set of 13 lung oncofetal miRNAs. Cancer-associated pathways targeted by the oncofetal miRNAs were identified, and clinical relevance was probed through their association with LUAD patient survival. Furthermore, this large cohort of whole miRNA transcriptome sequences of human fetal lung tissue provides a unique resource for the study of developmental and oncogenic processes.

## 2. Materials and Methods

### 2.1. Cohort Composition

The discovery (BCWH) cohort comprises three sample groups: ANL (*n* = 77), LUAD (*n* = 63), and FL (*n* = 25). The ANL and LUAD samples within the BCWH cohort were previously collected at Vancouver General Hospital under informed, written consent of the patients and approval by the University of British Columbia/BC Cancer Agency Research Ethics Board (H15-03060). Tumour section histology was reviewed and microdissected to >80% tumour cell content by a lung pathologist. Twenty-four of the FL samples were obtained as anonymous pathological autopsy specimens, from the Children’s and Women’s Pathology laboratory following ethics approval from the University of British Columbia and the Children’s and Women’s Health Centre of BC (H06-70085). This tissue was obtained from chromosomally normal (46XX/46XY) fetuses in their second trimester (17–24 weeks gestation), after elective terminations for medical reasons. Total RNA for the remaining FL sample was obtained from Biocat GmbH (Heidelberg, Germany). TRIzol reagent (Life Technologies, Carlsbad, CA, USA/ThermoFisher, Waltham, MA, USA) was used to isolate total RNA from frozen fetal and lung tissue samples. The total RNA was sequenced using the Illumina HiSeq2000 platform (Illumina Inc., San Diego, CA, USA) to obtain miRNA sequencing data. Samples with <5 million reads were excluded from our data set. MiRNA expression data for FL, ANL and LUAD samples have been deposited in the NCBI Gene Expression Omnibus (GSE175462) [19].

For validation, we obtained miRNA sequencing data from The Cancer Genome Atlas (TCGA) data repository, containing ANL (*n* = 38) and LUAD (*n* = 389) samples (dbGaP Project ID: 6208; https://portal.gdc.cancer.gov/ (accessed on 11 July 2019)). Samples with low coverage (<5 million reads) were omitted from this study. Corresponding clinical information was downloaded from the University of California Santa Cruz (UCSC) Xena Browser (http://xena.ucsc.edu/ (accessed on 11 July 2019)) (Table 1).

### 2.2. Quantification of miRNA Expression

Small RNA sequencing files were obtained as Binary Alignment Map (BAM) files and converted to unaligned FASTQ files. These were processed using the online platform miRMaster (July 2019), under default parameters [20]. MiRMaster trimmed adapters, performed quality filtering, and aligned sequences to the hg38 assembly. Novel miRNA prediction was also performed by miRMaster, which uses the AdaBoost algorithm, trained on a group of features that include measures of free energy and folding, to predict whether unannotated sequences are likely to represent genuine miRNA precursors [20]. Finally, miRMaster quantified expression of both annotated and previously unannotated miRNAs. Following miRMaster analysis, we averaged the reads of TCGA patient samples that had triplicate tumour sampling. For each cohort, miRNAs that had ≥1 read per million (RPM) in ≥10% of all samples in a given sample group (LUAD, ANL, or FL) were considered to be expressed in that group.

### 2.3. Assessment of Sample-Type Specificity

To visually assess the differences between the miRNA expression profiles of different sample groups, t-distributed Stochastic Neighbour Embedding (t-SNE) was used to cluster lung tissue samples according to their expression levels of all expressed miRNAs (*Partek^®^ Flow^®^* software, Partek Incorporated, St. Louis, MO, USA. 

### 2.4. Characterization of Differential and Oncofetal Expression

In order for a miRNA to be considered differentially expressed between any two sample types in the BCWH cohort, an average fold change (FC) >2 and post-hoc *p*-value < 0.05, as determined by ANOVA, were required. A miRNA was considered differentially expressed between the TCGA ANL and TCGA LUAD samples if it had FC > 2 and BH-adjusted *p*-value < 0.05, as determined by an unpaired *t*-test. A miRNA was considered to be oncofetal if it was: (a) overexpressed in both the BCWH LUAD and BCWH FL samples relative to BCWH ANL, (b) overexpressed in the TCGA LUAD samples relative to TCGA ANL, and (c) not expressed (<10% of samples with RPM ≥ 1) in both the BCWH ANL and TCGA ANL groups.

### 2.5. Prediction of Oncofetal miRNA Targets

Target genes of lung oncofetal miRNAs were predicted using mirDIP (v. 4.1.11.2; http://ophid.utoronto.ca/mirDIP, accessed on 28 May 2021), which integrates predictions from 30 distinct miRNA-target prediction tools [21]. Only the most confident 1% of predictions (those receiving ‘Very High’ integrative scores) were considered. Pathway enrichment analysis of target genes was performed using pathDIP (v. 4.0.21.4; http://ophid.utoronto.ca/pathDIP, accessed on 28 May 2021), which draws data from 22 distinct major databases of human biological pathways [22]. Extended pathway associations were considered, with only experimentally-confirmed protein-protein interactions, and a minimum confidence level of 0.99 for predicted associations. Subsequent word enrichment analysis was also performed using pathDIP (v. 4.0.21.4).

### 2.6. Assessment of Oncofetal miRNAs as Cancer Diagnostic Markers

Expression levels of lung oncofetal miRNAs in the BCWH cohort were used to train a support vector machine (SVM) classifier to identify the samples as either malignant or non-malignant. This miRNA-based classifier was then applied to the expression data from the TCGA LUAD and ANL samples. Normalized miRNA expression values for the TCGA Pan-Cancer cohort, including both malignant and non-malignant samples, were downloaded from the UCSC Xena Browser. Samples within the Pan-Cancer cohort were grouped by tissue of origin and used to test the diagnostic value of the lung oncofetal miRNA-based SVM in other tissues.

### 2.7. Association of Oncofetal miRNAs with Survival

Survival data for patients in the TCGA cohort were also collected from the UCSC Xena Browser and were available for 379 of the 389 TCGA LUAD patients. Follow-up information was also available for 58 of the 63 BCWH LUAD patients. Each patient was scored as positive (RPM ≥ 1) or negative (RPM < 1) for each individual oncofetal miRNA. A univariate log-rank test (*p* < 0.05) was performed to assess the association between the expression status of each oncofetal miRNA and patient outcome.

### 2.8. Software for Statistical Analysis and Illustrations

All statistical analysis and illustrations were performed with SPSS (v. 21.0; SPSS, Chicago, IL, USA), BRB-ArrayTools (v. 4.4.0; https://brb.nci.nih.gov/BRB-ArrayTools/, accessed on 28 May 2021) and GraphPad Prism (v. 5.0; GraphPad Software Inc., La Jolla, CA, USA) software.

## 3. Results

### 3.1. Novel miRNA Discovery and miRNA Profiling in Lung Tissues

A total of 423 miRBase v22-annotated (‘known’) miRNAs were expressed in at least one sample group within both the BCWH and TCGA cohorts. A further 219 unannotated (‘novel’) miRNAs were predicted in one or more of the BCWH cohort sample groups. Of these 219 novel miRNAs, 44 were also predicted in at least one TGCA cohort sample group, increasing the likelihood that these were indeed novel miRNAs and not sequencing artifacts (Appendix A) [23]. Downstream analysis considered only the 467 miRNAs detected in both the BCWH and TCGA cohorts (Figure 1A). Clustering of lung tissue samples by their expression of these miRNAs shows near complete separation of the samples by their origin (ANL, LUAD, or FL), demonstrating sample-type specificity (Figure 1B).

### 3.2. Differential Expression of miRNAs in Lung Adenocarcinoma

As many miRNAs are known to be deregulated in cancer, we compared the expression levels of the detected miRNAs between LUAD and ANL in both cohorts (Appendix A). We observed 26 underexpressed miRNAs and 145 overexpressed miRNAs in LUAD as compared to ANL in both cohorts (Figure 1C). These include two underexpressed and 12 overexpressed novel miRNAs (Appendix A).

### 3.3. Congruence between miRNA Expression in Lung Adenocarcinoma and the Fetal Lung

Out of the 26 miRNAs that were underexpressed in LUAD, 10 were also underexpressed in FL relative to ANL (Appendix A). Two of these miRNAs, including one novel miRNA (working ID: LUAD-nov-miR-43, mapping to chr22-:40956705-40956768), were not expressed in LUAD or FL.

These miRNAs could conceivably function to limit some of the behaviours that are common in tumours and fetal tissue but absent from healthy adult tissue, such as rapid cell division. Out of the 145 miRNAs that were overexpressed in LUAD, 70 were also overexpressed in FL relative to ANL (Figure 2A). Overall, there is a significant similarity between the LUAD vs. ANL differential expression status (overexpressed, underexpressed, or not significantly altered) of a miRNA and its FL vs. ANL status, demonstrating a congruence between the LUAD and FL tissues (Fisher’s exact test, *p* = 5 × 10^−9^).

Thirteen of the miRNAs that were overexpressed in both LUAD and FL were not expressed in ANL samples (Figure 2A). We denoted these miRNAs as lung oncofetal miRNAs, as they are expressed during fetal lung development, are not expressed in the adult lung, and their expression is reactivated during tumourigenesis (Table 2). Remarkably, five of these 13 sequences map to the imprinted C14MC miRNA cluster at the 14q32 band (Figure 2B, Table 2). Additionally, one novel miRNA (working ID: LUAD-nov-miR-26) that was overexpressed in both LUAD and FL relative to ANL also mapped near the C14MC cluster (chr14+:103337845-103337905).

### 3.4. Lung Oncofetal miRNAs Target Cancer-Related Pathways

In order to investigate the potential functions of these 13 lung oncofetal miRNAs, they were subjected to an mRNA target prediction analysis, which yielded 373 mRNAs that were predicted to be targeted by at least three of the 13 miRNAs (Appendix A). These 373 mRNA targets were enriched for members of a number of pathways related to cancer, including multiple databases’ entries for the Wnt signaling pathway (Table 3). Performing word enrichment analysis on the names of all significantly enriched pathways (with a false discovery rate < 0.05) confirmed that ‘Wnt’ is one of the most significantly enriched non-generic terms (Appendix A). The predicted involvement of these 13 oncofetal miRNAs in developmental and cancer-associated pathways, together with the upregulation of these miRNAs in fetal and tumour samples, suggests that they might work to activate these pathways through downregulating pathway repressors.

### 3.5. Lung Oncofetal miRNAs Hold Promise as Biomarkers of Non-Small Cell Lung Cancer

As lung oncofetal miRNAs are expressed in LUAD but not in normal adult lung tissue, we sought to investigate their utility as biomarkers of LUAD. After being trained on ANL and LUAD samples from the BCWH cohort (Figure 3A), a SVM was used to classify samples from the TCGA Pan-Cancer cohort as either non-malignant or malignant, based on their expression of lung oncofetal miRNAs. The classification of LUAD samples was highly accurate (AUC = 0.963) (Figure 3B,C). LUAD is the major subtype of non-small cell lung cancer (NSCLC), and we observed that the SVM was also able to classify samples of the other common type of NSCLC, lung squamous cell carcinoma (LUSC), with an AUC of 0.985 (Figure 3B,C). Additionally, the largely-correct classification of uterine, bladder, and head and neck samples (AUC > 0.9) suggests that these lung oncofetal miRNAs may also play a role in general cancer-related processes.

### 3.6. Lung Oncofetal miRNA Expression Is Inversely Correlated with Survival

In order to assess the prognostic utility of these oncofetal miRNAs, we looked at whether their expression levels were associated with the survival outcomes of LUAD patients. In the TCGA cohort, the overexpression of three individual oncofetal miRNAs (hsa-miR-329, hsa-miR-380, and hsa-miR-1290) showed significant association (*p* < 0.05) with a decrease in overall survival (Figure 4). However, no significant associations were observed in the BCWH cohort (Appendix A). Interestingly, these three miRNAs share a number of predicted mRNA targets (Appendix A), including *TRIM33*, which is an E3 ubiquitin ligase and an established tumour suppressor gene. *TRIM33* has been shown to ubiquitinate nuclear β-catenin, thereby inhibiting the canonical Wnt signaling pathway [25].

## 4. Discussion

We have quantified miRNA expression within 25 fetal lung samples, as well as within lung adenocarcinoma and adult non-neoplastic lung samples from two independent cohorts. In total, 467 miRNAs were expressed in at least one sample group of both the BCWH and TCGA cohorts. Of these miRNAs, 145 (31%) were overexpressed in LUAD tumours relative to ANL and only 26 (6%) were underexpressed. A significant fraction of these miRNAs were also differentially expressed between FL and ANL, most often in the same direction as their differential expression in LUAD. The substantial number of miRNAs that are deregulated similarly in LUAD and FL suggests that lung tumours often make use of fetal epigenetic regulators, such as miRNAs, in their appropriation of developmental pathways.

Our study has been limited to only miRNAs that are differentially expressed in lung adenocarcinoma. Other studies have documented that miRNA dysregulation also occurs in different types of lung cancer, and that some of the differentially expressed miRNAs are detectable in lung cancer patient serum (recently summarized by Zhong et al.) [26]. Therefore, the approach for identifying oncofetal LUAD miRNAs can be readily applied to other types of lung cancer. Of note, our study has been limited to the analysis of only fetal lung samples from the second trimester of gestational age; characterization of the miRNA transcriptome of earlier stages of lung development could lead to the identification of additional lung oncofetal miRNAs.

While previous studies have used microarrays to quantify the expression of sets of miRNAs in the fetal lung [27], an advantage of the transcriptome-based approach we take is the ability to identify novel miRNA sequences. In the course of analyzing the sequencing data for the BCWH and TCGA cohorts, we observed that 44 previously unannotated miRNAs were expressed in at least one sample group of each cohort. Studies such as this one that investigate miRNA expression in large numbers of samples from individual tissues have frequently uncovered novel miRNAs that were overlooked by multi-tissue miRNA discovery efforts [28,29]. The discovery of these 44 miRNAs expands the known lung miRNA transcriptome, providing new candidate regulatory molecules. Fourteen of these novel miRNAs were differentially expressed between ANL and LUAD, suggesting that they could possess cancer-related functions. Of particular interest is the discovery of a novel miRNA near the C14MC miRNA cluster that was overexpressed in LUAD and FL relative to ANL, and of a novel miRNA that was not detected in either LUAD or FL, but was expressed in ANL.

We identified 13 miRNAs that were overexpressed in both LUAD and FL but not expressed in ANL and denoted them as lung oncofetal miRNAs. Of these 13 miRNAs, five originate from the imprinted (maternally expressed) C14MC miRNA cluster at chromosome band 14q32. Deregulation of this miRNA cluster, which is located in the *DLK-DIO3* locus, has been linked to both fetal/placental developmental disorders and malignancy in adults [30,31]. Expression of miRNAs from this cluster has previously been associated with poor outcomes in LUAD, likely due to the promotion of cell migration and the epithelial-to-mesenchymal transition [32,33]. Hsa-miR-323b, which we identified as oncofetal in this study, is among the C14MC miRNAs that have shown this correlation [32]. The enrichment of oncofetal miRNAs within this cluster is consistent with the premise that they function in both developmental and tumourigenic pathways. Changes in methylation at differentially methylated regions (DMRs) within the *DLK-DIO3* locus, including the *MEG3* and intergenic (IG) DMRs, have been linked to dysregulated C14MC expression [32,34]. Further investigation of the regulation of C14MC miRNAs may lead to the discovery of additional fetal epigenetic regulators that are co-opted by tumours.

We observed that computationally predicted targets of the lung oncofetal miRNAs were enriched for genes involved in a number of developmental and cancer-related pathways, including the Wnt signaling pathway. Consistent with this, numerous past studies have experimentally demonstrated the downregulation of Wnt pathway repressors by specific lung oncofetal miRNAs. Hsa-miR-543 has been shown to downregulate *DKK1* [35], *WIF1* [36], and *SMAD7* [37], and to increase Wnt pathway activation [34]. MiRNA-mediated inhibition has also been shown between the miRNA/mRNA pairs hsa-miR-301b/*GPC5* [38], hsa-miR-433/*DKK1* [39], and hsa-miR-1290/*GSK3B* [40]. However, some of these miRNAs have also been implicated in targeting positive regulators of the Wnt pathway, such as *HMGA2* [41] and *SMAD2* [42]. Future study of these miRNAs may clarify their regulatory impact on the Wnt pathway and indicate whether that impact is responsible for the association between expression of certain lung oncofetal miRNAs and poor survival.

Numerous lung oncofetal miRNAs have also been previously implicated in Wnt-independent mechanisms that drive tumour-promoting phenotypes. For example, hsa-miR-1290 has been shown to target *SOCS4* and has been associated with heightened proliferation and invasive capacity in lung adenocarcinoma cell lines [43]. Similarly, hsa-miR-301b has been shown to target *BIM* and has been linked to increased proliferation and decreased rates of apoptosis in lung cancer cell lines [44]. Both hsa-miR-1290 and hsa-miR-301b have been shown to increase tumour size in cell line-derived xenograft mouse models [43,44]. Further supporting the potential functional significance of oncofetal miRNAs, we observed that the expression of three individual miRNAs, including two from the C14MC cluster, was negatively correlated with patient survival in the TCGA cohort. However, in the BCWH cohort we did not observe a significant negative correlation between overall survival and any of the oncofetal miRNAs. This could be due to the much smaller sample size of the BCWH cohort, which was composed of only 58 patients, in comparison to the 379 in the TCGA cohort.

We also found that expression levels of the 13 lung oncofetal miRNAs could effectively distinguish both LUAD and LUSC tumours from non-malignant tissues. The consistent lack of expression of these oncofetal miRNAs in healthy adult lung tissue makes them ideal candidate biomarkers for early detection or monitoring of lung adenocarcinoma. If the upregulation of these miRNAs is also observable in the serum of patients with early-stage NSCLC, then a lung oncofetal miRNA-based classifier could potentially be effective when applied to liquid biopsy samples. Should this prove possible, it would greatly expand the potential clinical utility of the classifier, as liquid biopsies are a relatively non-invasive, low-complication, and inexpensive modality of cancer screening [45]. However, as the SVM classifier constructed in this study accurately identified multiple types of cancer, the incorporation of additional lung-specific miRNAs might be necessary to optimize its specificity for NSCLC.

The accuracy of the classifier also underscores the minimal expression of the 13 oncofetal miRNAs in the healthy adult lung, which makes them candidate targets for the future development of anti-miRNA therapies. Antisense nucleotides are perhaps the most promising form of anti-miRNA therapy and have demonstrated efficacy in vivo in inhibiting the tumourigenic effects of certain miRNAs [46].

## 5. Conclusions

The inclusion of 25 human fetal lung samples in this study makes it the largest-scale whole miRNA transcriptome comparison of human fetal lung (FL) and adult lung (ANL and LUAD) samples to date. Data generated from this project’s fetal samples are a unique resource that will enable novel analyses of the fetal lung miRNA transcriptome in various contexts. The identification of 13 lung oncofetal miRNAs is a first example of the utility of these data. These miRNAs are predicted to interface with both developmental and cancer-related pathways, which is further evidence that lung tumours directly co-opt fetal epigenetic regulators. Subsequent studies may lead to an improved understanding of the mechanisms through which oncofetal miRNAs contribute to developmental disorders and tumour development, as well as the construction of refined diagnostic or prognostic miRNA panels.

## Figures and Tables

**Figure 1 cancers-13-02686-f001:**
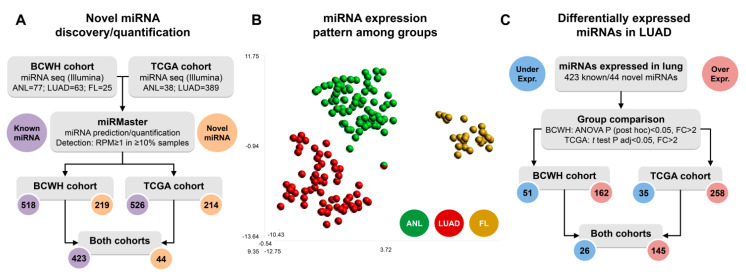
MiRNA expression profiles differ between lung tissue types. (**A**) Flowchart demonstrating the detection of known miRNAs and the prediction of novel miRNAs that were used in all downstream analysis; (**B**) t-SNE plot of the BCWH cohort samples, derived from expression of the 467 detected miRNAs; (**C**) Flowchart indicating the number of miRNAs differentially expressed in LUAD vs. ANL in each cohort.

**Figure 2 cancers-13-02686-f002:**
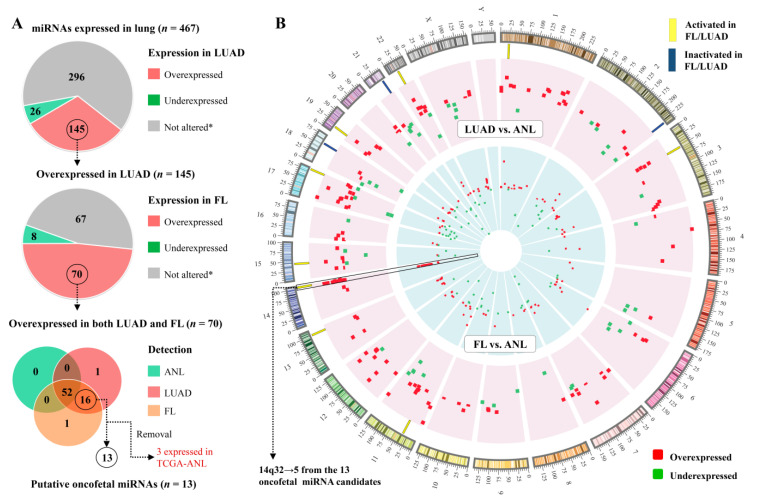
Many miRNAs display an oncofetal expression pattern. (**A**) Pie charts and Venn diagram illustrating the criteria for classifying miRNAs as oncofetal. * Statistically significant differences in expression were not observed in both datasets (BCWH and TCGA), or the sequence did not meet the expression threshold (RPM ≥ 1 in ≥10% of samples) in the indicated sample group; (**B**) Circos plot [24] of the genomic locations of miRNAs that were identified as differentially expressed in the discovery (BCWH) cohort in LUAD vs. ANL (outer ring, representing 162 overexpressed and 51 underexpressed miRNAs), or in FL vs. ANL (inner ring, representing 152 overexpressed and 70 underexpressed miRNAs). Oncofetal miRNAs are indicated by yellow rectangles.

**Figure 3 cancers-13-02686-f003:**
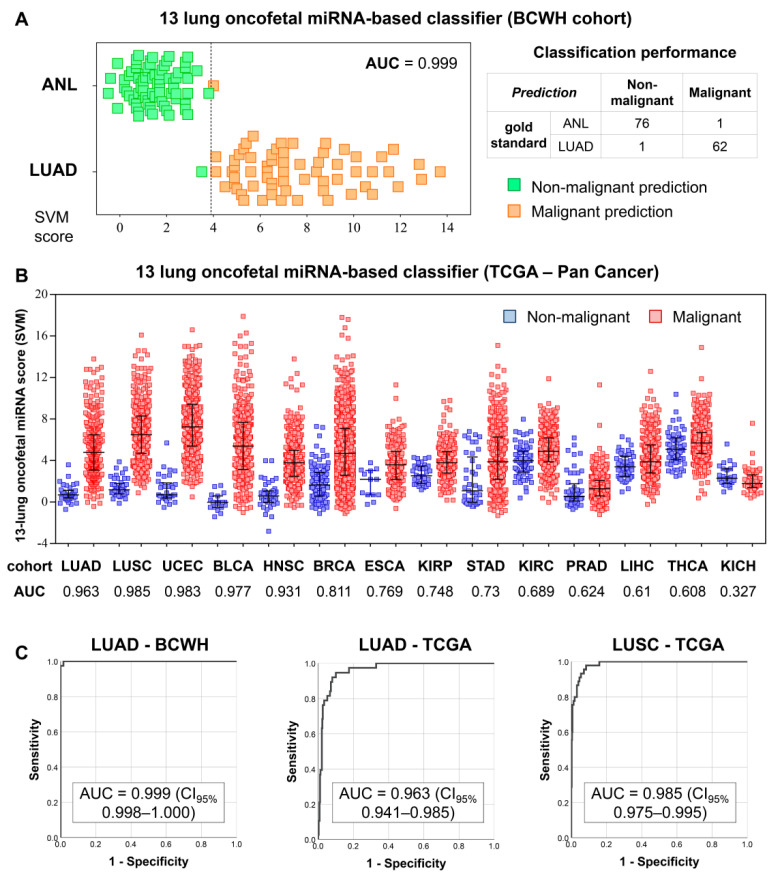
Training and testing of a lung oncofetal miRNA-based SVM. (**A**) Results of the training of the SVM on the BCWH cohort. Dotted line indicates the cut-off value of the SVM score. The formula for the SVM score was: Score = (1.24 × miR-301b) + (0.86 × miR-323b) + (−0.47 × miR-329) + (−0.24 × miR-380) + (−0.42 × miR-433) + (−0.26 × miR-543) + (0.01 × miR-627) + (1.12 × miR-6516) + (0.44 × miR-1290) + (0.34 × miR-1343) + (1.01 × miR-3170) + (0.97 × miR-4787) + (0.02 × miR-5684), with all expression values in units of log_2_(RPM). AUC = area under the receiver operating characteristic curve; (**B**) Distributions of the SVM scores for the non-malignant and malignant samples within the TCGA Pan-Cancer cohort. Error bars indicate the mean score ± standard deviation. LUSC: lung squamous cell carcinoma, UCEC: uterine corpus endometrial carcinoma, BLCA: bladder urothelial carcinoma, HNSC: head and neck squamous cell carcinoma, BRCA: breast invasive carcinoma, ESCA: esophageal carcinoma, KIRP: kidney renal papillary cell carcinoma, STAD: stomach adenocarcinoma, KIRC: kidney renal clear cell carcinoma, PRAD: prostate adenocarcinoma, LIHC: liver hepatocellular carcinoma, THCA: thyroid carcinoma, KICH: kidney chromophobe; (**C**) Receiver operating characteristic curves illustrating the performance of the SVM on the training (LUAD-BCWH) cohort, as well as on the LUAD-TCGA and LUSC-TCGA cohorts. CI_95%_: 95% confidence interval.

**Figure 4 cancers-13-02686-f004:**
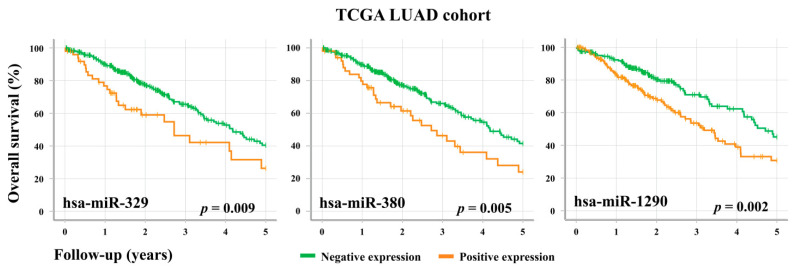
Overall survival of LUAD patients is associated with the expression levels of specific oncofetal miRNAs. TCGA LUAD patients (*n* = 379) were stratified according to their level of expression of the oncofetal miRNAs hsa-miR-329, hsa-miR-380, and hsa-miR-1290. For an individual miRNA, RPM < 1 was considered to be negative expression, while RPM ≥ 1 was positive expression.

**Table 1 cancers-13-02686-t001:** Clinical information for LUAD patients within the TCGA and BCWH cohorts.

Characteristics	TCGA (*n* = 389)	BCWH (*n* = 63)
Median Age (range)	66 (39–88)	70 (45–86)
Sex		
Male	173 (44%)	19 (30%)
Female	216 (56%)	44 (70%)
**Smoking Status**		
Current	83 (21%)	22 (35%)
Former	224 (58%)	18 (29%)
Never	64 (16%)	23 (37%)
**Stage**		
IA	102 (26%)	23 (37%)
IB	103 (26%)	18 (29%)
IIA	43 (11%)	2 (3%)
IIB	52 (13%)	11 (17%)
IIIA	57 (15%)	4 (6%)
IIIB	6 (2%)	1 (2%)
IV	17 (4%)	1 (2%)

**Table 2 cancers-13-02686-t002:** Names, genomic locations, and mean expression values of the 13 lung oncofetal miRNAs.

miRNA	Genomic Location	BCWH FL Mean ^1^	BCWH ANL Mean ^1^	TCGA ANL Mean ^1^	BCWH LUAD Mean ^1^	TCGA LUAD Mean ^1^
hsa-miR-1290	chr1−:18897078-18897096	5.86	0.06	0.31	1.10	7.86
hsa-miR-1343	chr11+:34941851-34941872	1.45	0.48	0.20	1.14	0.64
hsa-miR-301b	chr22+:21652990-21653011	5.82	0.28	0.10	5.49	2.13
hsa-miR-3170	chr13+:98208533-98208554	1.11	0.41	0.11	1.24	1.03
hsa-miR-323b	chr14+:101056233-101056255	7.41	0.34	0.12	13.15	7.73
hsa-miR-329	chr14+:101026797-101026819	9.05	0.50	0.30	1.62	0.81
hsa-miR-380	chr14+:101025021-101025042	4.54	0.23	0.33	1.35	0.93
hsa-miR-433	chr14+:100881897-100881918	12.07	0.52	0.24	2.62	1.38
hsa-miR-4787	chr3+:50675093-50675114	1.05	0.27	0.28	1.88	0.75
hsa-miR-543	chr14+:101032033-101032054	6.58	0.33	0.17	2.28	0.96
hsa-miR-5684	chr19+:12787132-12787151	0.67	0.09	0.16	0.45	0.55
hsa-miR-627	chr15-:42199630-42199651	1.20	0.52	0.49	1.12	1.15
hsa-miR-6516	chr17+:77089428-77089449	0.72	0.32	0.26	1.53	0.55

^1^ All mean expression values are given in units of reads per million (RPM).

**Table 3 cancers-13-02686-t003:** Predicted targets of lung oncofetal miRNAs are enriched for members of developmental and cancer-related pathways.

Pathway Database	Pathway ^1^	*p*-Value	FDR
NetPath	EGFR1	6 × 10^−10^	9 × 10^−7^
REACTOME	Transcriptional Regulation by TP53	4 × 10^−10^	1 × 10^−6^
REACTOME	Signal Transduction	2 × 10^−9^	2 × 10^−6^
ACSN2	G2_M_CHECKPOINT	4 × 10^−9^	3 × 10^−6^
NetPath	AndrogenReceptor	5 × 10^−9^	3 × 10^−6^
NetPath	Alpha6Beta4Integrin	8 × 10^−9^	3 × 10^−6^
KEGG	Wnt signaling	7 × 10^−9^	3 × 10^−6^
ACSN2	HEDGEHOG	1 × 10^−8^	5 × 10^−6^
Spike	WNT signaling	2 × 10^−8^	6 × 10^−6^
WikiPathways	Pathways Affected in Adenoid Cystic Carcinoma	2 × 10^−8^	6 × 10^−6^

^1^ The ten pathways with the lowest false discovery rates (FDRs) are shown. MiRNA target prediction analysis conducted using mirDIP v. 4.1.11.2. Pathway enrichment analysis performed by pathDIP v. 4.0.21.4.

## Data Availability

The data presented in this study for the BCWH cohort are openly available in the Gene Expression Omnibus at https://identifiers.org/geo:GSE175462, accessed on 28 May 2021, reference number [19].

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
