# Peer review of "Reactivation of Multiple Fetal miRNAs in Lung Adenocarcinoma"

_cancers, 2021, doi:10.3390/cancers13112686_

Round 1
Reviewer 1 Report
In their work, Cohn et al. using small RNA seq identified 13 miRNAs that were expressed in both lung adenocarcinoma tissue and fetal lung samples and are potential biomarkers. In their previous study (Becker-Santos et al. 2016) identified 37 miRNAs with common expression in human fetal lung tissue and NSCLC using a similar approach. Thus, the novelty of the study is not very high. However, the study by Becker-Santos et al. was performed on a small number of fetal tissues (n=4) comapred to Cohn et al. (n=25).
The introduction section is written very well and presents the background of the study. Materials and methods are described adequately.
Major points:
1. Authors identified 26 downregulated and 145 upregulated miRNA in LUAD comapred to ANL. However, the number is significantly substiantially lower compared to the results of a recent systematic review (Zhong et al. Translational Research 2021). Authors should at least discuss this discrepancy.
2. Authors state that "The absence of these microRNAs from non-malignant adult tissues also makes them ideal targets for novel therapies". However, Fig 3B shows that multiple types of healthy adult tissue express this oncofetal miRNAs! Thus, authors have to modified this sentence and be more careful with the interpretation of the results.
3. In current version, the study focuses only on bioinformatical analysis of the miRNA expression. To confirm the role of identified oncofetal miRNAs authors at least should try to demonstrate their role in the regulation of lung cancer cell behaviour (proliferation, invasiveness etc.).
4. Authors should present the ID and/or sequence of 13 identified oncofetal miRNA in the main text / figures / tables.
5. Authors should present the formula to calculate the miRNA score (SVM)
Minor points
L123 - Authors did not reported GSE number.
Authors should discuss the limitations of the study.
Reviewer 2 Report
The manuscript “Reactivation of multiple fetal miRNAs in lung adenocarcinoma” by David E. Cohn et al addresses an interesting and relevant concept concerning the association between cancer progression and the expression of a subset of miRNAs in human fetal lung samples and lung adenocarcinoma. The detection of such oncofetal microRNAs may be used as a tool for the identification of new markers and the development of innovative therapies in lung adenocarcinoma.
The results are well presented and well written and the conclusion is supported by results. In particular, the authors showed interesting data reporting the upregulation of 13 oncofetal miRNAs in tumor samples and their involvement in the activation of cancer-associated pathways. There are few comments:
- In figure 4 the authors reported a significant association between the overexpression of miRNAs (has-mir-329, has-mir-380, has-mir-1290) and a decrease in overall survival of patients. What types of pathways are modulated from such miRNAs? The authors should add this information in the text.
- The authors should add in the discussion some comments about the potential development of innovative therapies by modulation of oncofetal miRNAs activities in lung adenocarcinoma.
Reviewer 3 Report
The present manuscript titled “Reactivation of multiple fetal miRNAs in lung adenocarcinoma” by Cohn DE et al utilized a combination of RNA-seq followed by bioinformatic-platform to analyze and identify a unique signature of 13 oncofetal miRNAs that are expressed both in the fetal-lung as well as in lung adenocarcinomas, however, not in the adult normal lung. They have concluded this novel cohort of 13 miRNAs to be “Reactivated” in lung cancer after probably being silent in the course of normal lung development and maturation. And at this point, this reviewer has a strong argument that needs to be clarified by the authors.
- The authors did not pursue any functional studies for these unique 13 miRNA signatures in either of the systems; then how did the authors confident about the term “Reactivation” as a functional reactivation?
- In continuation of the previous question, these 13 annotated and verified (by the wet lab sequencing data) miRNAs need to be listed in a separate table mentioning their functional relevance, target mRNA or other molecules, signaling pathways involved/interfering in, etc., based on the published literature both from the fetal as well as, oncogenic perspective with proper references. This is very important to assemble for the general readers and to justify the ‘Reactivation’ terminology in the title.
- 1A, C is illegible to some extent, either make them bigger by using a bigger font size or change the color of the blue/green boxes used in the schematics.
Reviewer 4 Report
As a reviewer, I enjoyed reading this article. It is well-written logically, and the authors clearly demonstrated that several miRNAs are associated with cancer development and abnormal fetal and placental development, thus potentially serving as biomarkers for cancer detection. The authors utilized the fetal lungs, but I think the ethical considerations were fully described in the manuscript.
I could suggest just a few points to improve the manuscript.
Minor concerns:
- In Figure 3, ROC curve should be shown in LUAD and LUSC.
- In the Discussion section, the authors should describe the limitation of the study and clinical applicability of the findings. The classifier may not be specific to lung cancer cases, since it worked for the other cancer types (UCEC, BLCA, HNSC, and so on) Liquid biopsy, specifically designed for the detection of lung cancer, may be the next step.
